# Glypican-3 (GPC-3) Structural Analysis and Cargo in Serum Small Extracellular Vesicles of Hepatocellular Carcinoma Patients

**DOI:** 10.3390/ijms241310922

**Published:** 2023-06-30

**Authors:** Montalbano Mauro, Perricone Ugo, Zachary Walton, Shirafkan Ali, Cristiana Rastellini, Luca Cicalese

**Affiliations:** 1Department of Neurology, University of Texas Medical Branch, Galveston, TX 77555-5302, USA; mamontal@utmb.edu; 2Molecular Informatics Group, Fondazione Ri.MED., 90133 Palermo, Italy; uperricone@fondazionerimed.com; 3John Sealy School of Medicine, University of Texas Medical Branch, Galveston, TX 77555-5302, USA; zcwalton@utmb.edu; 4Rutgers Health, Department of Cardiac Surgery, New Brunswick, NJ 08901, USA; Ali.Shirafkan@rwjbh.org; 5Department of Surgery, University of Texas Medical Branch, Galveston, TX 77555-5302, USA; crrastel@utmb.edu; 6Department of Neurobiology, University of Texas Medical Branch, Galveston, TX 77555-5302, USA

**Keywords:** hepatocellular carcinoma, Glypican-3, small extracellular vesicle, serum marker

## Abstract

Glypican-3 (GPC-3) is a heparin sulfate proteoglycan located extracellularly and anchored to the cell membrane of transformed hepatocytes. GPC-3 is not expressed in normal or cirrhotic liver tissue but is overexpressed in hepatocellular carcinoma (HCC). Because of this, GPC-3 is one of the most important emerging immunotargets for treatment and as an early detection marker of HCC. To determine if GPC-3 domains associated with serum small extracellular vesicles (sEVs) could be used as an HCC diagnostic marker, we predicted in silico GPC-3 structural properties and tested for the presence of its full-length form and/or cleaved domains in serum sEVs isolated from patients with HCC. Structural analysis revealed that the Furin cleavage site of GPC-3 is exposed and readily accessible, suggesting the facilitation of GPC-3 cleavage events. Upon isolation of sEVs from both hepatocytes, culture media and serum of patients with HCC were studied for GPC-3 content. This data suggests that Furin-dependent GPC-3 cleaved domains could be a powerful tool for detection of initial stages of HCC and serve as a predictor for disease prognosis.

## 1. Introduction

Glypican-3 (GPC-3) is a cell surface-bound oncofetal proteoglycan which has been identified as a potential prognostic factor and immunotherapeutic target in hepatocellular carcinoma (HCC) [1], lung carcinoma, severe pneumonia, and acute respiratory distress syndrome (ARDS) [2]. GPC-3 is normally expressed in the fetal liver but not in healthy adult hepatocytes. Moreover, GPC-3 has been used as a target for molecular imaging and therapeutic intervention in HCC due to its elevated expression during cancerous transformation events [3,4,5]. To date, GPC-3-targeted magnetic resonance imaging, positron emission tomography, and near-infrared imaging have been investigated for early HCC detection [6,7]. Various immunotherapeutic protocols targeting GPC-3 have been developed using CAR-T cell technology [8], including the use of humanized anti-GPC-3 cytotoxic antibodies [9], treatment with peptide/DNA vaccines [10], immunotoxin therapies [11], and genetic therapies. In this study, we reported our findings regarding the structure, function, and biology of GPC-3 with a focus on its clinical potential as a diagnostic marker in HCC patient serum.

Furin is a cellular endoprotease and convertase that activates many precursor proteins in secretory pathway compartments. Furin plays a crucial role in different cellular processes and is involved in many diseases, including cancer [12,13]. Furin is a type 1 transmembrane serine endoprotease (794 amino acids—94 kDa) belonging to the proprotein convertase family that is expressed in all vertebrates and many invertebrates [14,15,16]. It is localized and sorted according to its cytoplasmic domain by the trans-Golgi network/endosomal system: a structure responsible for sorting and packaging proteins for transport to their final destinations such as the cell surface, endosomes, lysosomes, and secretory granules [17,18]. Furthermore, Furin serves a key role in allowing researchers to understand the regulation of protein trafficking in mammalian cells. Furin cleaves GPC-3 (R358/S359) into a 30 kDa C-terminal domain and a 40 kDa N-terminal domain [19,20]. Recently, it has been shown that inhibition of GPC-3 cleavage results in a reduced proliferation of the HepG2 cell line [21]. Although the biological significance of GPC-3′s cleaved products is unclear, studies suggest that they play a role in hedgehog protein signaling [20,22]. No evidence of signaling functions mediated by cleaved forms of GPC-3 associated with extracellular vesicles (EVs) has been reported previously.

(EVs), including small extracellular vesicles (sEVs), are phospholipid membrane-enclosed nanoscale vesicles. Following MISEV2018 nomenclature from the International Society of Extracellular vesicles [23], any EV with a diameter <200 nm is referred to as a small extracellular vesicle. In general, these vesicles have been proposed to act as cargo carriers or mediators of intercellular communication. These vesicles are also thought to support neoplastic growth and dissemination of messengers mediating tumor-stroma and tumor-tumor communication [24] while simultaneously inducing oncogenic mutations in nearby cells. Vesicles involved in oncogenic stimulation are further known as oncosomes [25]. Considering their cellular membrane location, it is not surprising that glypican proteins are a component of the secretory pathway as GPC-1 has been observed in pancreatic and breast cancers [26]. GPC-3 cleavage products were first observed in the serum of patients with HCC in 2003 [27]. Interestingly, Hippo et al. in 2004 identified an N-terminal soluble form of GPC-3 in HCC serum, which was elevated in HCC compared to cirrhotic and healthy patients [28]. More recently, EVs emerged as new promising liquid-biopsy biomarkers for cancer diagnosis. Di H. et al. used a multiplex profiling assay (NAISA, nanozyme-assisted immunosorbent assay) to evaluate with high accuracy the presence of GPC-3 in HCC along with EV proteins, such as CD63 (cluster of differentiation 63), CEA (carcinoembryonic antigen), PD-L1 (programmed cell death ligand (1) and HER2 (human epidermal growth factor receptor (2) [29]. The scarcity of research pertaining to serum GPC-3 provides impetus for exploring the occurrence of GPC-3 within serum-derived sEVs. In this study, we will refer to the serum vesicles isolated as small extracellular vesicles (sEVs) based on the most recent nomenclature approved by MISEV18 report [23] from the International Society of Extracellular Vesicles (https://www.isev.org/) accessed on 21 April 2023. Nomenclature is based on the size analysis performed with Nanosigth and based on the current classification [30].

In clinical practice, the high rate of HCC local recurrence suggests the presence of different hepatocyte populations within the liver and particularly in proximity to the tumor. We investigated primary human hepatocyte cultures obtained from liver specimens of patients affected by cirrhosis and HCC and further observed cultures for signs of characteristic proliferation and/or transformation. In this study, liver samples were obtained from HCC cirrhotic patients immediately after surgery. Cell outgrowth and primary cultures were obtained from the HCC lesion, the cirrhotic tissue proximal (CP, 1–3 cm) and distal (CD, >5 cm) to the margin of the neoplastic lesion. Cells were kept in culture for 16 weeks. We observed that HCC cells maintained their morphology and unmodified neoplastic characteristics when cultured. Cells isolated from CP showed a progressive morphologic transformation into HCC-like cells and were accompanied by signs of invasiveness.

Due to the high rate of HCC recurrence and its increased incidence rate in the US and worldwide, it is becoming crucial to identify biomarkers from liquid biopsy that can improve diagnosis and prognosis of HCC. To this end, we identified higher levels of GPC-3 domains in HCC serum sEVs in the cohort considered. Moreover, we observed an oncogenic function of conditioned media from primary human hepatocytes (PHH) isolated from HCC and a critical function of Furin-dependent GPC-3 cleavage in cancer cell viability and GPC-3 domain accumulation in sEVs. This study is intended to fuel the interest of clinicians and researchers to investigate sEVs as biomarkers in a larger cohort of patients with HCC, cirrhosis, and healthy individuals, which was limited in number in the present study.

## 2. Results

### 2.1. GPC-3 Structure Shows a Stable Conformation with Furin Cleavage Site Exposed to Solvent

Molecular Dynamics simulations were conducted for 500 ns on a protein model based on the Crystal structure of GPC-3 (PDBid 7AZW). Missing residues were fixed through homology modeling techniques. Analyzing simulation data, Root Mean Squared Deviation (RMSD) showed that GPC-3′s structure stabilized within the first 20 ns and maintained its stability for the entire simulation: indicating that it did not undergo large conformational changes. As depicted in the Root Mean Squared Fluctuation plot (RMSF), the major contributions to protein RMSD were given by the C and N-terminus domains, with a certain mobility of the unfolded region where the Furin cleavage site is placed. The protein stability was also confirmed by energy analysis. From the energy plot, indeed, the trend is stable with a smooth decrease. Root Mean Square Fluctuation (RMSF) and Root Mean Squared Deviation for the protein backbone are plotted in Figure 1a,b. Energy analysis plot is depicted in Figure 1c.

Analyzing the tridimensional structure of GPC-3 (Figure 1d) the Furin-cleavage site (Arg358-Ser359) appears clearly exposed at the solvent accessible area throughout the simulation, suggesting it is an accessible site for the Furin enzyme. Figure 1e shows Solvent Accessible Surface Area (SASA) for the cleavage site. As reported in the literature [31], the average value for the two amino acids (Arg358-Ser359) should be around 223 Å for Ser and 335 Å for Arg. As observable in the SASA plot in Figure 1e, the average area of the two residues remained high throughout the simulation, demonstrating that their position is perfectly exposed to the solvent and is available for contact with other proteins (e.g., Furin).

### 2.2. C-Terminal GPC-3 Domain Is Localized in Hepatocyte Small EVs

To study the effects of CD-, CP- and HCC hepatocyte-secreted components in normal (NL) hepatocytes (Hep), which are not migratory cells, we implemented a no-contact co- culture utilizing a multichambered system (graphically represented in Figure 2a). Normal hepatocytes were plated in the upper chamber coated with Matrigel, and CD-, CP- and HCC hepatocytes were plated in the lower chamber. Representative phase contrast images of the lower chamber (left column) and Matrigel chamber (upper chamber) with migrating hepatocytes revealed cells migrating through the Matrigel (Figure 2b). Quantification of the area (expressed in % of covered area by migrating cells) covered by migrating cells in NL-Hep/CD-Hep, NL-Hep/CP-Hep and NL-Hep/HCC systems suggested that the materials exchanged between the chambers induce migration of normal hepatocytes (Figure 2c). In particular, we observed a significant increase in the number of migrating NL-Hep induced by CD-Hep at 12 h and CP-Hep at 24 h. This observation, alongside our previous studies on CP-Hep as pre-cancerous cells, sets our experimental plan focusing on CP-Hep. Immunofluorescence assay in CP-Hep was performed to show the cellular distribution of GPC-3 C-terminal (GPC-3_C_) and N-terminal domains (GPC-3_N_). To evaluate co-occurrence of GPC-3 domains in CD63 positive small EVs we measure Manders’ colocalization coefficient (MCC). While the GPC-3_N_ is detected in the cytosol and membrane of hepatocytes (Figure 2d) with a partial overlapping with CD63 (enriched in cellular vesicles) in the central region of the cells, GPC-3_C_ shows a higher co-localization and MCC with CD63 compared to GPC-3_N_ (Figure 2e). Moreover, GPC-3_C_ showed minimal reactivity in the cytosol and cellular membrane. Co-localization with CD63, an EVs marker, also revealed that not all CD63 positive sEVs contained GPC-3_C_ (Figure 2d,e). To confirm our IF observations, we performed Western blotting analysis of cytoplasmic and sEV associated content of GPC-3 domains. We observed GPC-3_N_ in the cytosol extract with a lower level of another form modified by a post-translational event in the sEVs (Figure 2g). Another form of GPC-3 between 20–25 kDa is detected in initial stages of culture EVs and disappears in late stages. In Figure 2f we observed that GPC-3_C_ is, as confirmed by immunofluorescence, unique to small EVs and is entirely absent from the cytosol. This form is maintained over time and shows a lower molecular weight than GPC-3_N_. To show the purity of the extraction, GAPDH was used as a control for the cytosolic extract and Hsp70 for the small EVs isolated extract. Densitometry analysis as a function of Hsp70 showed higher levels of GPC-3_N_ in HCC sEVs (Figure 2h). GPC-3_C_ densitometry at early (1 week) and late (8 weeks) passages showed higher levels in Cp-Hep and HCC small EVs (Figure 2i,j) We know that GPC-3 is cleaved by Furin-Convertase, and that the domains are still associated by disulfide bindings [32]. However, in the small EVs, GPC-3_C_ expression is higher than GPC-3_N_, which suggests that the C domain is the main form of GPC-3 in secreted sEVs from PHH. In Figure 2k is represented the merged membrane of immunoblots represented in Figure 2f,g.

### 2.3. The GPC-3 C-Terminal Domain Product from Proteolytic Furin-Convertase Induced Cleavage Is Localized in Hepatocyte Small EVs

To test Furin proteolytic activity on GPC-3, we incubated CP-Hep for 24 h with different concentrations of Furin I Inhibitor, CMK. Hepatocytes from 10 to 100 µM showed a change in morphology characterized by cytoplasmic shrinkage and loss of adhesion with the substrate (Figure 3a). Hepatocytes showed a CMK concentration dependent Anoikis (Figure 3a, white arrows): a form of cellular apoptosis induced by the inability of normal cells to attach to the extracellular matrix (ECM). Incorrect attachment, or lack of ECM attachment, could affect cell growth and differentiation in normal cells. However, research studies have determined that metastatic cells can undergo Anoikis without affecting their ability to migrate and invade surrounding or distant organs. Moreover, we quantified the number of dead cells using trypan blue assay, observing a concentration-dependent toxic effect of CMK treatment (Figure 3b). To evaluate the effects of CMK on GPC-3 domain levels, we performed a Western Blot (WB) of Cytosolic and small EV protein extracts (Figure 3c). We used GAPDH and Hsp90 as loading controls for cytoplasm and small EV extracts, respectively. We observed that the Furin enzyme is strongly associated with sEV extracts, and that it is gradually present in its inactivated form in a concentration-dependent fashion upon exposure to CMK. No significant bands of Furin were detected in the cytosolic protein extract. Interestingly, GPC-3 showed a remarkable reduction of cleaved forms: confirming it is a Furin protein substrate in hepatocytes. Moreover, GPC-3_C_ is detected in sEVs while GPC-3_N_ is observed in the cytoplasm of CP-Hep. This finding suggests that after cleavage by Furin, GPC-3_N_ translocates to the cytoplasm to exert an unknown function not investigated here. De Cat B. et al. showed that GPC-3 processing by Furin-like enzyme is necessary to regulate Wnt signaling and cell survival [32]. Further, GPC-3 processing is required to modulate hedgehog protein signaling [20]. A recent study revealed that inhibition of GPC-3 cleavage results in reduced cell proliferation in liver cancer cell lines [21], which is also confirmed by our observations. All this evidence supports our findings, thus suggesting that Furin activity and GPC-3 cleaved products are important for cancerous cell survival. IB quantification confirmed that CMK dose-dependent inhibition of Furin (Figure 3f) is associated with significant reduction of cleaved C- and N-terminal domains of GPC-3 at 25, 50 and 100 µM CMK (Figure 3d,e, respectively). To evaluate effects of CMK on cell viability, we performed a Co-IF of Cleaved Caspase 3 and GPC-3_N_ (Figure 3g) in CP-Hep (0 vs 100 µM CMK)**.** As observed in phase contrast images, hepatocytes exposed to CMK showed remarkable shrinkage of the membrane with a compact IF signal of GPC-3_N._ Cleave Caspase 3 was also observed with high IF signal compared to untreated hepatocytes (Figure 3g—right panels). To validate single frame observation, we took images with the confocal microscope and used Imaris software to measure the area (µm^2^), volume (µm^3^), and number of cells containing GPC-3_N_. In Appendix A, we show the 3D reconstruction of CP-Hep untreated and treated with CMK (top 10×, bottom 60×). GPC-3 in untreated cells showed a wide distribution of fluorescent signal, while CMK cells showed shrinkage and clustering. CMK treatment significantly reduced the area and volume of hepatocytes (Appendix A, respectively). Z-Stack and orthogonal view confirmed a flat and homogenous distribution of cellular volume (Appendix A) while CMK induces a reduction of cellular volume and a higher condensation of GPC-3N in the periphery of the cells which undergo anoikis (Appendix A).

### 2.4. Glypican-3 Is Present in Human Serum sEVs

To evaluate GPC-3 presence in human serum sEVs, we collected blood samples from 10 HCC patients before surgical removal of HCC lesions. We then isolated sEVs from each patient’s serum using an established protocol described in material and method Section 4.6. Schematic representation of sEVs extraction is represented in Figure 4a, pellets were used for size and morphology characterization using Nano Sight NS3000 and Transmission Electron Microscopy (TEM), respectively. Protein contents were evaluated with western blot. Representative TEM image of sEVs isolated from HCC serum are reported (Figure 4b, scale bar: 0.1 µm) Evaluation of 3 controls and 6 HCC cases via Western Blot identified sEVs containing two GPC-3 domains recognized by individual antibodies (Ab): one that recognized the C-terminal domain, and one that recognized the N-terminal domain (Figure 4c,d). In GPC-3_C_ immunoblotting (Figure 4c), we detected a single band around MW of 25 kDa in both control and HCC samples. In detail, we observed that the band from HCC cases appeared higher and thicker than control samples. The GPC-3_N_ Ab recognized a 3-band pattern (Figure 4c): a 75 kDa band (representing the full length, uncleaved GPC-3), a band around 50 kDa, and a third band around 25 kDa. Moreover, we evaluated GPC-3 content in soluble fraction of HCC and control sera (Figure 4e). GPC-3 (N-terminal Ab) immunoblot showed that GPC-3 serum content is associated with small EVs, and its N-terminal domain is poorly observed in soluble fraction of patient sera (Figure 4f).

## 3. Discussion

Here, we lay the foundation for further investigation involving higher numbers of samples and cases to evaluate GPC-3 content in small EVs isolated from HCC serum. We observed that cell media components (including small EVs) can induce neoplastic properties in normal hepatocytes. Further characterization revealed a key role of Furin-dependent cleavage in cancer cell survival, suggesting that the cleavage of GPC-3 is crucial for cell proliferation, as also observed by [21,28,33]. Furin produces two GPC-3 domains, one N-terminal cytoplasmic domain and one C-terminal domain concentrated in sEVs. These findings suggest that each domain serves a different function in the cell not investigated here.

We showed, for the first time in primary human hepatocytes, that GPC-3 exhibits a complicated regulation of its domains, particularly its N- and C-terminal. Different cancer cell lines produce GPC-3 [2] but we established PHH cultures with peculiar characteristics. This different distribution confirmed by western blotting opens a new view of the molecular biology of GPC-3 in transformed hepatocytes. We showed previously that Furin-Convertase co-localizes centrally within cells with the standard form of GPC-3 found in the TGN (Trans Golgi Network) or endosomal system where it is cleaved into R358/S359. We also showed that the cleavage site is exposed on the protein surface to enzymatic cleavage by Furin. Inhibition of Furin with CMK showed that GPC-3 cleavage is important for cancer cell survival as it generates the GPC-3_C_ form observed in the sEVs of PHH.

GPC-3 is primarily concentrated in sEVs with a small representation of its soluble component in the serum. In human serum, GPC-3 sEVs are represented by one form of 25 kDa GPC-3_C_ and two forms of GPC-3_N_ at 40 and 30 kDa. Comparative analysis of sEV content in HCC patients with healthy controls showed a slightly higher band in HCC patients suggesting a slightly higher molecular weight form of GPC-3. Two major post translational modifications (PTMs) of GPC-3 have been studied in the carcinogenic context: glycosylation and cleavage [22,34]. The current opinion is that extensive PTM dysregulation leads to the pathogenesis of HCC [34]. GPC-3 has two heparan sulfate chain insertion sites that mediate most interactions of the GPC-3 protein. The grade and pattern of sulfonation seems to modulate interactions and function of the heparan sulfate side chain [35,36]. Also, GPC-3 is anchored to the membrane thanks to glycosylphosphatidylinositol anchor insertion, the absence of this anchor induces a soluble form of GPC-3 that differentially modulates HCC proliferation [22]. In general, the study of GPC-3 PTMs in the cells as well as in sEVs presents an exciting opportunity for drug development. Interestingly, there is not a linear correlation between N- and C-forms as is seen in PHH GPC-3 sEVs. This finding indicates that GPC-3 travels into blood circulation as a group of cleaved forms rather than its whole, uncleaved form. The variation in N- and C-form concentrations indicates differential function on target tissues.

In this study, we evaluated GPC-3 in serum small EVs from HCC patients. These findings suggest that GPC-3 is a candidate as a serum tumor marker for diagnostic and prognostic evaluation of HCC. Moreover, furin inhibition was observed with reduction of pre-cancerous cell viability offering it as a target for HCC therapy. Further, this study has some limitations due to the number of samples analyzed. Future studies will include samples from patients with cirrhosis (no HCC) and healthy liver (no cirrhosis, no HCC, no HCV). A longitudinal study will also be relevant in determining GPC-3 levels in small EVs before and after resection/ablation of HCC lesions to study GPC-3 as a prognostic marker.

## 4. Materials and Methods

### 4.1. Missing Residues Homology Modelling

The full structure of the GPC-3 protein is not currently available in the PDB database, www.rcsb.org [37]. So, starting from available substructures (PDBid: 7ZAV, 7ZAW) we used homology modelling techniques to add missing residues, obtain a full 3D model, and submit it to molecular dynamics. The process of sequence finding and protein building was set up using the online software Swissmodel [38], which is a reliable software widely used for homology modelling techniques. From the models obtained using Swissmodel, we considered the one with the highest GMQE (Global Model Quality Estimation) and [39]. The obtained structure was refined and optimized with Maestro suite 21.3 (Schrödinger Release 2022-1: Maestro, version 13.1, Schrödinger, LLC, New York, NY, USA, 2021) to correct atom clashes, bond order and H-bond networks. The resulting structure was analyzed with the software *PropKa,* and the protonation states of amino acids were assigned under the assumption of a solution pH of 7.4 [40,41].

### 4.2. Molecular Dynamics (MD)

The optimized model was inserted into an orthorhombic water box containing salt to neutralize charges, and the pH was set to 7.4 using Maestro—Desmond software (DESMOND: Schrödinger Release 2022-1 version 13.1, Schrödinger, LLC, New York, NY, USA, 2021). The system was subjected to an initial minimization protocol. During this process, the system stabilized temperature and pressure. Once the system had been minimized and equilibrated at a temperature of 310 K with pressure of 1 atm, the system proceeded with the molecular dynamics production. The simulation was conducted for 500 ns using OPLS4 Force Field and TIP3P water model. Nose-Hoover chain Thermosat with relaxation time of 1 ps was set. The Barostat chosen was the Martyna-Tobias-Klein with a relaxation time of 2 ps and an isotropic coupling. The simulation starting point was set with a random seed.

### 4.3. Patients and Samples Collection

Patients were enrolled following Institutional Review Board approval and informed consent in accordance with UTMB (University of Texas Medical Branch) institutional policies. All patients were diagnosed with primary HCC, and none had received any preoperative treatment. The patients underwent surgical resection, and serum samples were collected on the day of surgery. Histopathological evaluation of the tissue samples was performed using standard hematoxylin and eosin staining. Sections were examined by a pathologist to confirm diagnosis of HCC and to rule out neoplastic contamination in cirrhotic liver samples utilized as cirrhotic proximal (CP) and cirrhotic distal (CD) for the present study. Their clinic-pathological and demographic characteristics are presented in Appendix A. The protocol for collecting clinical samples was approved by the Ethics Committee of the University of Texas Medical Branch, and the patients provided written informed consent before samples were collected. Samples were obtained from 6 patients with liver cirrhosis, HCV+/HBV− and well-differentiated focal HCC undergoing liver resection (mean age 57). 3 patients with HCC-free livers from a pathological perspective were used as a control group. Fresh tissue samples were obtained at the time of surgery, immediately placed in cold (4 °C) sterile saline solution and processed. Histopathological evaluation of the tissue samples was performed using standard hematoxylin and eosin staining. Sections were examined by a pathologist to confirm HCC diagnosis and rule out neoplastic contamination in cirrhotic liver samples utilized as cirrhotic proximal (CP), cirrhotic distal (CD) and hepatocellular carcinoma (HCC) for this study. Additional samples from liver tissues (CP, CD and HCC) were individually placed in RNAlater Stabilization Reagent (Qiagen) and immediately cryopreserved at −160 °C.

### 4.4. Isolation and In-Vitro Culture of Primary Human Hepatocytes

Tissue specimens obtained were washed in phosphate buffer saline (PBS) and processed within 2 h of surgical resection. Samples were washed with physiologic solution, minced with fine sterile scissors and scalpeled into fragments of approximately 1 mm^3^. Cells were immediately isolated from cirrhotic tissue proximal (1 < CP < 3 cm from the tumor resection margin) and distal (CD > 5 cm from the tumor resection margin or contra-lateral lobe) to the HCC lesion. The cell isolation procedure was performed as previously shown [42]. Briefly, the 1 mm^3^ fragments of tissue were incubated for 3 h with Fetal Bovine Serum (FBS) HyClone (Fisher Scientific, Hampton, NH, USA). FBS was then replaced by complete RPMI 1640 medium with 10% FBS, 1% of antibiotics (Corning Inc., Corning, NY, USA) and amino acids (Sigma-Aldrich, St. Louis, MO, USA; MEM Non-essential amino acid solution (100×) #M7145) and incubated for 24 h. Every 48 h cells were then washed with 2 mL of RPMI 1640 complete medium. After 8 weeks a monolayer of primary cells around the explants was observed. Cells were detached using Trypsin/EDTA 1X (Corning), re-plated, and maintained in culture at 37 °C and 5% CO_2_.

### 4.5. Cell Culture and Furin Inhibitor Treatment

CP-Hep was treated with Furin Inhibitor I—(Calbiochem 344930 Sigma-Aldrich, St. Louis, MO, USA) and Decanoyl-RVKR-CMK at a final concentration of 10, 25, 50 and 100 µM for 24 h. After CMK exposure, bright-field pictures were acquired. Culture media and cell pellets were collected for protein extractions and western blotting. CMK cytotoxicity was measured using trypan blue assay.

### 4.6. Microvesicle Isolation from Culture Medium and Human Serum

Primary human hepatocytes were seeded in 6 well plates (10^6^ cells/well) and incubated the next day with new media. At 24 h post-isolation, the extracellular vesicle fraction was isolated from the culture medium (1 mL) using the Total Exosome Isolation kit (Invitrogen, Waltham, MA, USA #4478359). The extracellular vesicle pellet was lysed with SDS-sample buffer for Western analysis followed by determination of the protein concentration by BCA. In these experiments, we used exosome-depleted (A2720801 Thermo Fisher Scientific, Waltham, MA, USA) fetal bovine serum. Six HCC patients and three healthy people at the University of Texas Medical Branch were enrolled in this study.

### 4.7. Nanoparticle Tracking Analysis (NTA)

NTA measurements were performed for size and concentration of nanoparticles using a NanoSight NS300 instrument (NanoSight, Malvern Panalytical, Malvern, UK), following the manufacturer’s instructions. Medium of BEAS 2B cells were diluted to reach a particle concentration suitable for analysis with NTA (106 to 109 particles/mL). Two different dilutions for each sample were prepared and analyzed three times for each one. The samples were injected into the NS300 unit (approximately 300 µL) with a 1 mL sterile syringe. The capturing settings (shutter and gain) and analyzing settings were manually set according to the protocol as explained in the technical note and then optimized for specific nanoparticles. The NanoSight NS300 recorded 60 s sample videos which were then analyzed with the Nanoparticle Tracking Analysis (NTA) 2.0 Analytical software. The ize and concentration of PDF reports and videos were obtained for each set of data (5 fields recorded for each sample). The average of all 6 reads per sample are shown. Size profiles of small EVs isolated in this study are presented as Appendix A.

### 4.8. Immunofluorescence and Confocal Imaging

Cells were fixed with 3.7% formaldehyde (Sigma-Aldrich) for 10 min at room temperature (r.t.) and permeabilized with 0.1% Triton X-100 (Sigma-Aldrich) in PBS for 5 min. Cells were then rinsed and covered with PBS blocking buffer (1% BSA in PBS) for 30 min at 37 °C to minimize non-specific adsorption of the antibodies to the coverslips. After washing with PBS, cells were incubated with the primary antibodies (anti-GPC-3 and anti-α-SMA, Abcam, Cambridge United Kingdom; diluted in PBS + 1% BSA + 0.05% NaN_3_) at 4 °C overnight. Preparations were washed three times with PBS and incubated for 1 h at room temperature with secondary antibodies, either Alexa Fluor 488 (Abcam #150113) or Alexa Fluor 596 (Abcam #150080) diluted 1/1000 in 1% BSA + 0.05% NaN_3_. Nuclei were counterstained with 2.5 µg/mL Hoechst 33342 (NucBlue^®^ Live ReadyProbes^®^ Reagent; Life Technologies, Carlsbad, CA, USA #37605), for 15–20 min. Following three washes with PBS, cells were examined on Olympus BX51 optic microscope equipped with fluorescence and suitable filters for Alexa Fluor 488, Alexa Fluor 596 and DAPI detection; images were captured and photographed using a computer-imaging system (PictureFrame^TM^). Primary antibodies used for immunofluorescence staining included: Albumin (1:500, Santa Cruz (F-10) sc-271605), Cytokeratin18 (CK18, 1:100, Abcam ab9217), Heppar1 (1:100, DAKO M7158, Agilent Technologies, Santa Clara, CA, USA), CD68 (1:500, Santa Cruz USA, sc-393951), CD31 (1:500, Santa Cruz, sc-376764) and β-Catenin (1:250, Abcam ab32572). Manders’ Colocalization Coefficients (MCCs) was calculated using ImageJ FIJI software to measure co-occurrence of one protein that colocalizes with the other protein. MCCs are reported and presented as a bar graph to measure the co-occurrence of GPC-3 domains in small EVs.

### 4.9. Transmission Electron Microscopy (TEM)

After isolation as described in Section 4.6, the ultrastructure analysis of isolated small EVs was performed with 5 μL drops adsorbed on a 200-mesh coated resin grid (FCF 200—CU Formvar/Carbon, Electron Microscopy Sciences, Hatfield, PA, USA) for 10 min at room temperature (RT). Grids were blotted with filter paper and stained with 2% aqueous uranyl acetate (cat# 541-09-3, Electron Microscopy Sciences) for negative staining for 1 min at RT. The uranyl acetate was then removed using filter paper, and the grids were dried with warm regular light for 2 min. Images were acquired with a Philips CM-100 transmission electron microscope (UTMB Core) at 60 kV with an Orius SC2001 digital camera (Gatan, Pleasanton, CA, USA).

### 4.10. Protein Extraction and Western Blotting

Western blotting was performed on whole cell lysates to detect C-terminal GPC-3 (clone 1G12, sc-65,443, Santa Cruz Biotechnology, Inc. Dallas, TX, USA raised against a fragment containing amino acids 510–580 of GPC-3 of human origin) and N-terminal GPC-3 (clone 9C2 ab129381, Abcam, sequence recognized between aa 55–200) and Furin-Convertase (ab3467, Abcam). Cells were cultured and harvested before confluence. 1 × 10^7^ cells were lysed using a modified RIPA buffer [150 mM NaCl, 25 mMTris (pH 7.4), 1 mM EDTA, 1 mM EGTA, 2 mM Na_3_VO_4_, 10 mM NaF, 1% NP40, 10% glycerol, aprotinin (10 mg/mL) and leupeptin (10 mg/mL)]. Supernatant was collected and quantified by a BCA protein assay (Thermo Fisher Scientific)). Equal amounts of proteins were separated by SDS-PAGE and transferred to nitrocellulose membrane (LI-COR Biosciences Lincoln, NE, USA #926-31090), which was blocked using 5% non-fat dry milk in Tris-Buffered saline with Tween 20 (Blocking Buffer Li-Cor #927-40040). The membrane was incubated overnight at 4 °C with the primary antibodies listed above. After incubation, the membrane was washed 3 times with PBST and then rinsed and incubated for 1 h at r.t. in appropriate anti-mouse or anti-rabbit IRDye 680–800 secondary antibodies (Li-Cor Biosciences). The membrane was rinsed, developed with Odyssey Imaging Systems Li-Cor, and specific protein bands were detected with Image Studio Software (Version 4.0.21 Li-Cor). Hsp70, Hsp90 and GAPDH served as loading controls.

### 4.11. Indirect Co-Culture

To study the effects of the secretome in normal hepatocytes, we used a two-chamber system to expose normal hepatocytes to secretome of CD-, CP-, and HCC PHH. Briefly, we plated PHH from normal livers in the upper chamber and CD-, CP-, and HCC hepatocytes in the lower chamber. A Matrigel was placed between layers to allow for the exchange of secreted vesicles and biological materials between the two chambers. Observation of NL-Hep in the Matrigel is an indication that media from the lower chamber induces migration capacity that is not associated with normal hepatocytes but with transformed cells.

### 4.12. Statistical Analysis

All in vitro experiments were performed in triplicate at least. Micrographs from human cells were obtained from three independent experiments with comparable results, and all figures show representative micrographs. All data are presented as mean ± SD (standard deviation) and were analyzed using GraphPad Prism Software 9.0. Statistical analyses were carried out using Student’s *t* tests or one-way analyses of variance (ANOVAs) followed by Tukey’s multiple comparisons testing. Column means were compared with one-way ANOVA using treatment/condition as the independent variable. Group means were compared using two-way ANOVAs with factors on treatment. If ANOVA revealed a significant difference, pairwise comparisons between group means were performed with Tukey and Dunnett’s multiple comparison tests.

## Figures and Tables

**Figure 1 ijms-24-10922-f001:**
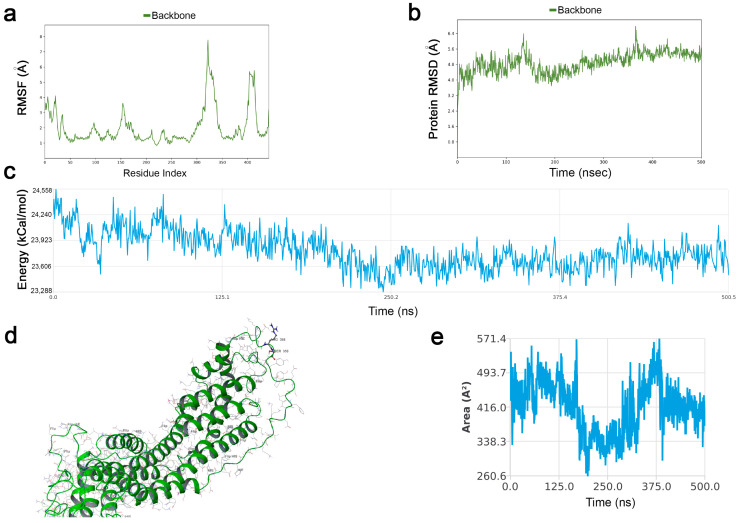
GPC-3 structure shows a stable conformation with Furin cleavage site exposed to solvent. (**a**) Protein Backbone Root Mean Squared Fluctuation. The plot demonstrates the conclusion that the main fluctuations are observed within the N and C terminus of the protein together with the unfolded region between residues 300–350. The Furin cleavage site (358–359) maintains a certain stability to fluctuations. (**b**) Protein Backbone Root Mean Squared Deviation. Here, the plot shows good overall backbone stability during the entire 500 ns simulation. Once stabilized (within the first 20 ns) there are no significant deviations, and there are no “jumps” in the RMSD trend. (**c**) Energy Analysis Plot. The global internal energy of the protein (Coulomb, van der Waals, bonds, angles, and dihedrals energy) was calculated and plotted. As shown in the picture, the global internal energy decreases during the simulation. In (**d**) The position of the Furin cleavage site is shown. The two residues are in an unfolded region accessible to Furin as also reported in the SASA plot (**e**) showing a high rate of solvent accessible surface area for this region.

**Figure 2 ijms-24-10922-f002:**
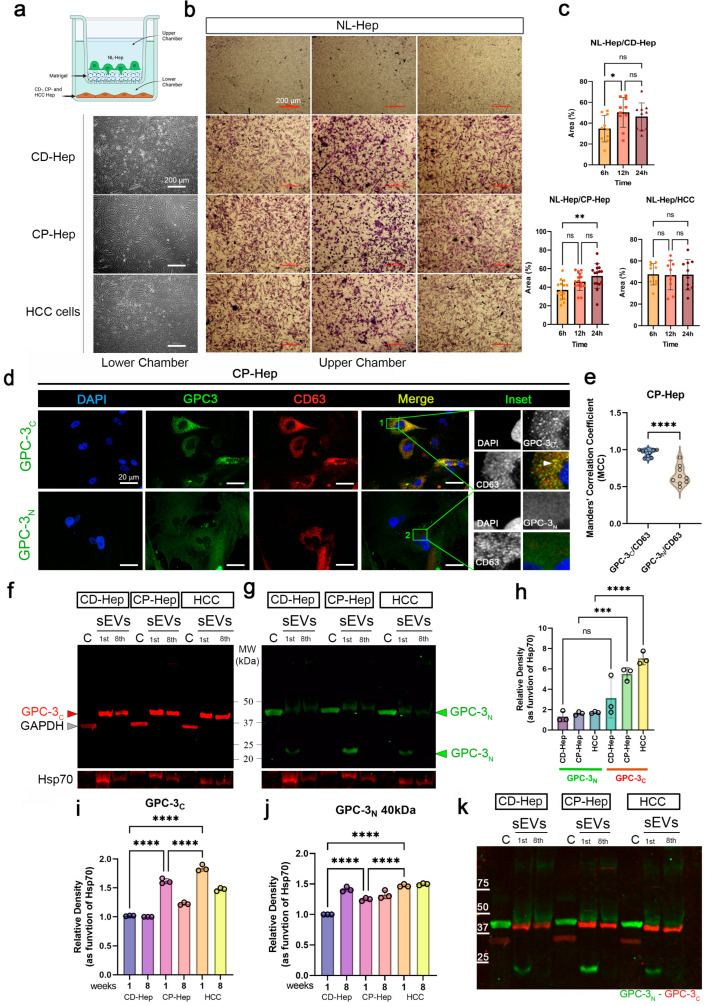
C-terminal GPC-3 domain is localized in hepatocyte small EVs. (**a**) schematic of the multichambered system. Normal hepatocytes plated in the upper chamber coated with Matrigel and CD-, CP- and HCC hepatocytes plated in the lower chamber. (**b**) Representative phase contrast images of the lower chamber (left column) and matrigel chamber (upper chamber) with migrating hepatocytes (magnification 4×, scale bar: 200 µm). (**c**) Quantification of the area (expressed in % of covered area) covered by migrating cells in NL-Hep/CD-Hep, NL-Hep/CP-Hep and NL-Hep/HCC systems. Ordinary one-way ANOVA and Tukey’s multiple comparison test was performed, *, *p* < 0.038 and **, *p* < 0.017. (**d**) Representative IF images of CP-Hep stained with DAPI-nuclei (blue). GPC-3_C_ and GPC-3_N_ (green), CD63 (red) and merged channels. Inset of magnified region of interests are reported (magnification 63×, scale bar: 20 µm). (**e**) Manders’ Colocalization Coefficients are reported to measure colocalization of GPC-3 domains in CD63 positive sEVs (unpaired *t*-test ****, *p* < 0.001). (**f**) Immunoblot of GPC-3_C_ and GPC-3_N_ (**g**) in sEVs isolated from CD-, CP- and HCC-Hep at 1st and 8th week of cultures, alongside cytoplasmic fractions. GAPDH, loading control for cytoplasm, Hsp70 as loading control for sEVs. (**h**) Relative density of GPC-3_N_ and GPC-3_C_ in sEVs fractions (Ordinary one-way ANOVA, Tukey’s multiple comparison test ***, *p* < 0.001 and, ****), (**i**) Relative density of GPC-3_C_ in sEVs fractions in 1 and 8 weeks (Ordinary one-way ANOVA, Tukey’s multiple comparison test, ****). (**j**) Relative density of GPC-3_N_ in sEVs fractions in 1 and 8 weeks (Ordinary one-way ANOVA, Tukey’s multiple comparison test, ****). (**k**) IB overlapped GPC-3_C_ (red) and GPC-3_N_ (green).

**Figure 3 ijms-24-10922-f003:**
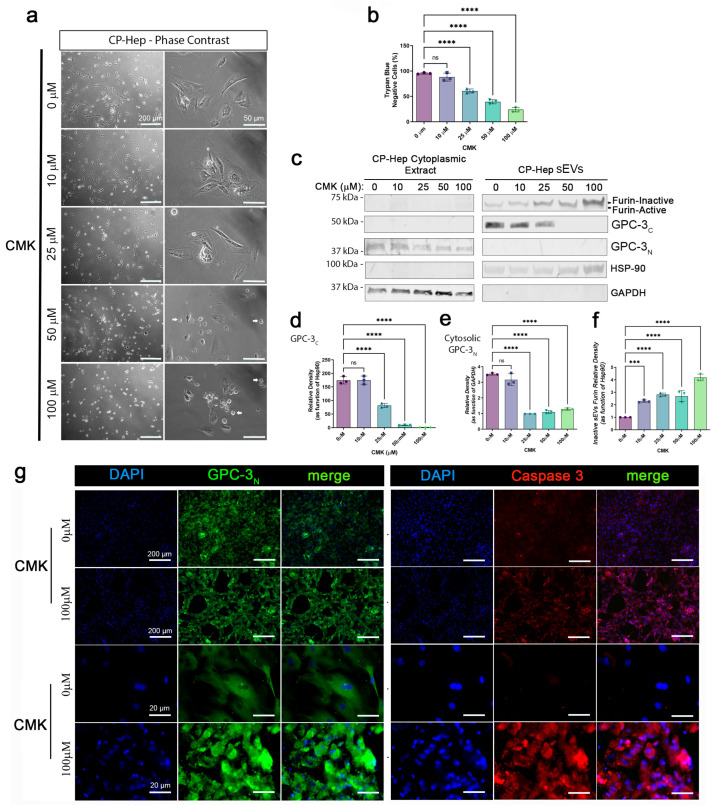
GPC-3_C_ is a product of proteolytic activity of Furin-Convertase and accumulates in cell-derived small EVs. (**a**) Representative phase contrast images of CP-Hep exposed to CMK (0, 10, 25, 50, 100 µM) scale bar: 200 µm (4×) and 50 µm (20×). (**b**) Cellular Viability expressed in the percentage Trypan Blue negative cells exposed to CMK (One-way ANOVA ****, *p* < 0.001). (**c**) WB of CP-Hep cytoplasmic and sEV protein fractions immunoblotted for Furin, GPC-3_C_, GPC-3_N_, HSP90 and GAPDH. (**d**) GPC-3_C_ densitometry in CP-Hep IB (**e**) Cytoplasmatic GPC-3_N_ densitometry in CP-Hep (**f**) Quantification of Inactive Furin in sEVs. (**g**) Representative Co-IF of GPC-3_N_ and Caspase 3 in untreated and treated (100 µM CMK) CP-Hep (scale bar: 200 µm (4×), 20 µm (63×). Statistical analysis in b, d, e, and f are performed with ordinary one-way ANOVA and Dunnett’s multiple comparison test, comparing the mean of each treated group with mean of control group, ***, *p* < 0.001 and ****, *p* < 0.0001, ns as non-significant are reported.

**Figure 4 ijms-24-10922-f004:**
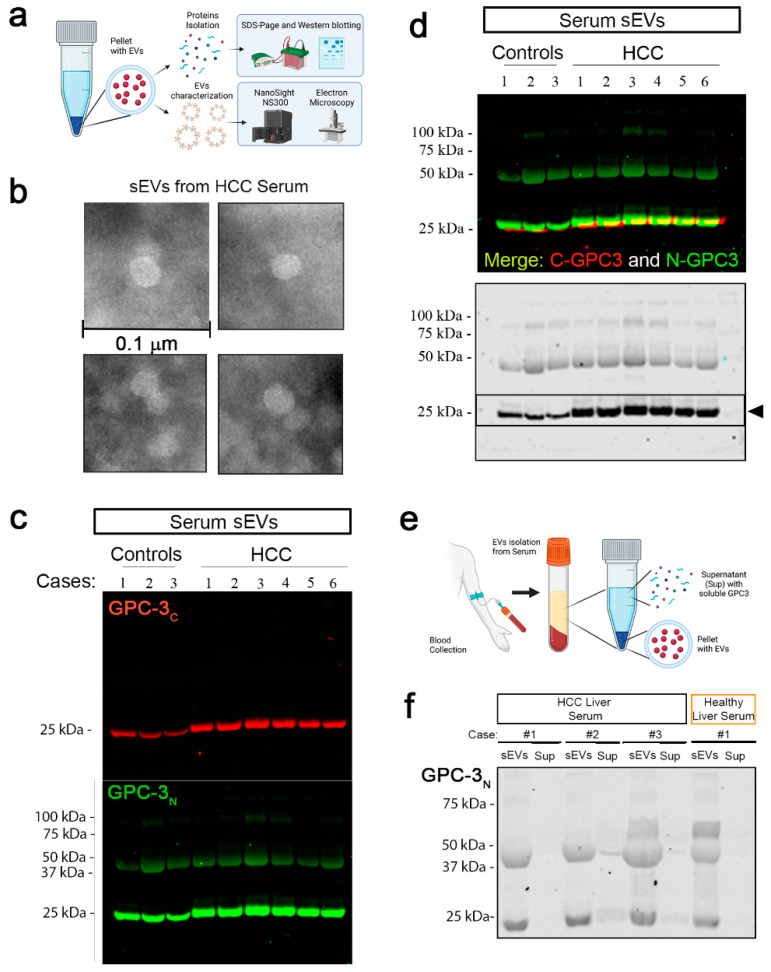
Glypican-3 in human serum small EVs. (**a**) Schematic representation of sEVs extraction. Pellets were used for size and morphology characterization using Nano Sight NS3000 and TEM, respectively. Protein contents were evaluated with western blot. (**b**) Representative TEM image of sEVs isolated from HCC serum (scale bar: 0.1 µm). (**c**) GPC-3 sEVs content was evaluated by WB in healthy controls and HCC cases. GPC-3_C_ (red-top panel) and GPC-3_N_ (green-bottom panel) antibodies were used for immunoblotting. (**d**) Upper blot—Merged immunoblot of N- and C-terminal GPC-3 IBs, Lower Blot—GPC-3_N_ IB is represented as well in black/white. (**e**) schematic representation for serum sEVs and supernatant isolation to verify GPC-3 soluble content. (**f**) IB of GPC-3_N_ in sEVs and corresponding supernatant (Sup) in three HCC cases and 1 control.

## Data Availability

Data available on request from the authors. The datasets used and/or analyzed in this current study are available from the corresponding author.

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
