# Peer review of "Glypican-3 (GPC-3) Structural Analysis and Cargo in Serum Small Extracellular Vesicles of Hepatocellular Carcinoma Patients"

_ijms, 2023, doi:10.3390/ijms241310922_

Round 1

Reviewer 1 Report

I read the manuscript entitled as “Glypican-3 (GPC-3) structural analysis and cargo in serum microvesicles of hepatocellular carcinoma patients” with great interest. Authors conducted structural and functional analyses on GPC-3 using human HCC. I think that this manuscript contained novel findings. However, I would like to confirm the followings to authors.

1.  In patients and sample collection section in materials and methods, you described ten HCC and 5 control samples. However, only six and three were used as HCC patient and control samples in Figure 5. Please explained the discrepancy.

2. In supplemental Table 1, you described pathologic characteristics of 10 HCC patients. The description is questionable. The relationship among tumor size, tumor focality, and TMN staging may be wrong. Also, all cases were diagnosed as well differentiated HCC. However, well differentiated HCCs are usually less than 2.0 cm. Well differentiated HCCs with 5.5 cm (case 6) or 7 cm (case 9) in a diameter were quite rare. Could you please submit the microphotographies of these cases? Furthermore, well differentiated HCC usually corresponds with Grade 1. However, you described that some cases demonstrate Grade 2 or 3 even in well differentiated HCC. Which classifications did you use in pathological diagnosis? Please confirm these findings to the institutional pathologists.

3. In this manuscript, authors used CP-Hep, CD-Hep, and HCC sample. Although you have been reported the differences of these previously, could you explain your previous data simply in the introduction section? I think additional description will be reader friendly.

4. I found some typos in your manuscript. Figure 4f (Line 203), Figure 4d (Line 205), Figure 4g (Line 206), Figure 4g (Line 209) and Figure 5a (Line 212) could be Figure 3f, Figure 3d, Figure 3g, Figure 3g, and Figure 4a, respectively. Please make sure all of your descriptions including these.

Author Response

Rebuttal Letter

We would like to thank the reviewers for their critiques and important considerations. We believe that the manuscript is significantly improved after we addressed all the major concerns.

Comments and Suggestions for Authors

Reviewer 1

I read the manuscript entitled as “Glypican-3 (GPC-3) structural analysis and cargo in serum microvesicles of hepatocellular carcinoma patients” with great interest. Authors conducted structural and functional analyses on GPC-3 using human HCC. I think that this manuscript contained novel findings. However, I would like to confirm the followings to authors.

Answer: We thank the reviewer for these important comments

  1. In patients and sample collection section in materials and methods, you described ten HCC and 5 control samples. However, only six and three were used as HCC patient and control samples in Figure 5. Please explained the discrepancy.

  Answer: Thank you, we corrected the discrepancy in patients and samples collection. In Supplementary Table 1 are presented 10 HCC cases used to isolate small EVs for consistency with Grade classification we use samples from 6 cases presenting a grade of differentiation G1.

  1. In supplemental Table 1, you described pathologic characteristics of 10 HCC patients. The description is questionable. The relationship among tumor size, tumor focality, and TMN staging may be wrong. Also, all cases were diagnosed as well differentiated HCC. However, well differentiated HCCs are usually less than 2.0 cm. Well differentiated HCCs with 5.5 cm (case 6) or 7 cm (case 9) in a diameter were quite rare. Could you please submit the microphotographies of these cases? Furthermore, well differentiated HCC usually corresponds with Grade 1. However, you described that some cases demonstrate Grade 2 or 3 even in well differentiated HCC. Which classifications did you use in pathological diagnosis? Please confirm these findings to the institutional pathologists.

Answer: Thank you for this important comment. All pathological classifications and size tumor have been evaluated and reported in each case report by the UTMB Pathologists and confirmed by our immunostaining and histological characterization as published previously1. We reported this note in the patient’s section.

  1. Montalbano M, Rastellini C, Wang X, Corsello T, Eltorky MA, Vento R and Cicalese L: Transformation of primary human hepatocytes in hepatocellular carcinoma. Int J Oncol 48: 1205-1217, 2016
  2. In this manuscript, authors used CP-Hep, CD-Hep, and HCC sample. Although you have been reported the differences of these previously, could you explain your previous data simply in the introduction section? I think additional description will be reader friendly.

 Answer: Thank you, we included in the introduction a paragraph where we described previous data. Line 86-96

  1. I found some typos in your manuscript. Figure 4f (Line 203), Figure 4d (Line 205), Figure 4g (Line 206), Figure 4g (Line 209) and Figure 5a (Line 212) could be Figure 3f, Figure 3d, Figure 3g, Figure 3g, and Figure 4a, respectively. Please make sure all of your descriptions including these.

Answer: Thank you and sorry for the typos, we corrected.

Reviewer 2 Report

Comments to Authors

The study by Mauro et al. aimed to analyze the expression of cleaved domains of Glypican-3 (GPC-3) and discussed the mechanism of cleavage by Furin. Also, the authors proposed that specific cleaved domains of GPC-3 expressed in serum microvesicles could be helpful as a potential biomarker in the early stages of HCC. However, although the general idea is quite attractive, the study has several drawbacks that affect the reliability of its findings.

-          One of the main concerns is the statistical analysis described in the paper. There are several sections where the authors could provide more detail and transparency to confirm the accuracy of their results. First, the number of samples and subjects should be included at least in the methods, which is also desirable in the results of each experiment. Second, it is unclear how the assumption of the normal distribution is confirmed, and non-normal distribution data is transformed to apply a parametric statistical test. Third, results from comparisons between the serum of patients are not presented; therefore, it is unclear how one-way ANOVA was used to analyze the relationship between GPC-3 domains in HCC and healthy patients, as described in the Statistical Analysis section. Also, one-way ANOVA was applied to compare the proteolytic activity of Furin-convertase (Fig. 3) but did not describe how multiple comparisons were tested. In addition, the authors should confirm if a post hoc analysis was conducted. Lastly, the statistical values of each experiment are desirable to include in the text or reported as supplementary data to the manuscript, such as exact p-values, error values, and confidence intervals. The authors are encouraged to provide more information on these points to improve the reliability of their findings.

Another area of concern is validity and interpreting the results.

-          The results from the first experiment reported in Fig.2a-c seem to be essential to understand why the authors continue investigations primarily on cirrhotic proximal (CP) samples in comparison to normal (NL) hepatocytes or cirrhotic distal (CD) or HCC cells. Otherwise, the authors not described or interpret these results in the manuscript and are only reported in Fig. 2. The following results to evaluate the proteolytic activity of Furin were conducted only in CP-Hep (Fig. 3 and 4). 

-          Complete information on antibodies should be fully described in the Methods. This is particularly important to understand how GPC-3 fraction N-terminal and C-terminal results are compared as one of the most relevant results of this study.

-          In this regard, another technique or statistical analysis should confirm the specificity of GPC-3 C-terminal expression in microvesicles. The authors proposed that the Pearson coefficient confirms the increase in the expression of C-terminal GPC-3. However, this statistical approach only demonstrates the linear relationship between two variables, in this case, the expression of GPC-3 and CD63 (a specific marker of microvesicles). Considering that this co-expression value does not confirm colocalization between two features is important. From the fluorescence images obtained, the authors could evaluate the relative fluorescence intensity resulting from colocalization (yellow) of GPC-3 (green) and CD63 (red); these results should be analyzed by Manders’ Overlap coefficient, taking into account the overlap between the two channels of fluorescence and provide a more accurate assessment of colocalization. Also, relative protein levels could be measured and compared between cytosol and purified microvesicles. The blots in Figure 2 suggest that a similar analysis was done but do not clearly show that this evaluation was performed. Fig. 2f and 2g show similar results, but only the expression of GPC-3 in microvesicles and GPC-3 N-terminal and C-terminal protein levels seem similar in 1st passage. Also, is confusing the results plotted in Fig.2h-j; it is not clear if the results are from the 1st or 8th passages and why this is relevant to this study. Given the relevance of this paper in discussing functional differences in N- and C-terminal fractions of GPC-3, authors should consider validating their results using another method, such as co-immunoprecipitation or proximity ligation assays.

-          Extracellular vesicles (EVs) and Microvesicles (MVs) terms are arbitrarily exchanged, which generates confusion on results. i.e., Figure 2f-g refers to MVs, but Fig.2k refers to EVs. According to the figure legend, these are the same results from hepatocytes. Therefore, it is necessary to adapt the terminology according to the current guidelines of the International Society of Extracellular Vesicles.

-          It is hard to understand the relevance of measuring the area and volume of cells GPC-3 N-terminal positive cells to the aim of the study. According to the previous results reported in this manuscript, GPC-3 C-terminal is the cleaved form present in microvesicles from CP-hepatocytes. It seems not to have (or not been discussed) a relationship with the altered area and volume of cells by Furin inhibition with CMK. As no relationship is probed between the inhibition of GPC-3 cleavage, its expression in microvesicles and its relation to the reported disturbances has not been established; it does not seem relevant to evaluate GPC-3c levels from serum vesicles as a possible biomarker for HCC.

-          In this sense, results from serum microvesicles in patients should be expanded to arrive at a more objective interpretation. The sample of 3 control subjects should be increased, and samples from cirrhotic patients would be included according to their previous results. Also, it is not clear the difference from immunoblots in Fig. 5c, 5d, and 5e; it seems that blots from 5c and 5d are the same, and differences should be reported quantitatively as in their previous results. In addition, blots from 5e are different in levels of protein of GPC-3 N-terminal. Lastly, if GPC-3 N-terminal is only expressed in the cytoplasm (Fig. 3C), this is an inconsistent result, whereas, in serum-purified microvesicles from patients, GPC-3 N-terminal is expressed (Fig. 5f).

Considering the authors’ need to review and confirm their results, their discussion should be expanded and contrasted with previous studies.

-          i.e., when the authors described that regulation of post-translational modifications was not evaluated in this study, authors should discuss if mechanisms that regulate GPC-3 post-translational changes were previously reported.

-          Authors should discuss why this post-translational modification described in HCC patients explains a physiopathological process associated with cancer progression. Without this, it is hard to understand how this finding could improve HCC diagnosis and prognosis.

-          Another point to discuss is what is previously reported about N- and C- cleaved forms of GPC-3 on target tissues in the context of cancer signaling mediated by microvesicles.

Finally, it is inaccurate to assume these results support the idea that Furin-dependent GPC-3 cleaved domains could be a powerful tool for detecting initial stages. Therefore, authors should be focused on validating the expression of cleaved forms in cell lines derived from HCC patients and different clinical conditions (cirrhotic and non-cirrhotic patients and different stages of HCC) to assure the validity of GPC-3 cleaved forms in the diagnosis and prognosis of HCC.

There are also minor comments that the authors should consider:

-          Citation style is not consistent (numbers than full cite)

-          There is information that needs to be properly cited (i.e., lines 56-57, 63-67, 69, 77-79)

-          ‘It is not surprising does not seem a sufficiently substantiated argument to suggest that GPC-3 could be part of a signaling mechanism mediated by microvesicles (Line 71-72)

-          Several references are duplicated.

-          Abbreviatures are first described in the Methods section; this may not be functional because this is the last section of the manuscript.

-          Figures are not correctly cited in the text. (i.e., line 134 describes the N-terminal of GPC-3 detected on cytosol and membrane of hepatocytes, referencing Fig. 2A. However, Fig. 2A is a schematic view of the multichambered system used to evaluate migration).

Overall, there are major areas to review in statistical methods, the validity and interpretation of results, discussion and citing, and minor comments about attending in general in the manuscript to conclude that on this initial state, the manuscript should be rejected to help the authors to produce a more impactful and reliable study.

Author Response

Rebuttal Letter

We would like to thank the reviewers for their critiques and important considerations. We believe that the manuscript is significantly improved after we addressed all the major concerns.

Reviewer 2

Comments and Suggestions for Authors

Comments to Authors

The study by Mauro et al. aimed to analyze the expression of cleaved domains of Glypican-3 (GPC-3) and discussed the mechanism of cleavage by Furin. Also, the authors proposed that specific cleaved domains of GPC-3 expressed in serum microvesicles could be helpful as a potential biomarker in the early stages of HCC. However, although the general idea is quite attractive, the study has several drawbacks that affect the reliability of its findings.

Answer: We thank the reviewer for these critical and constructive comments. We answered to the major concerns expressed.

-          One of the main concerns is the statistical analysis described in the paper. There are several sections where the authors could provide more detail and transparency to confirm the accuracy of their results. First, the number of samples and subjects should be included at least in the methods, which is also desirable in the results of each experiment. Second, it is unclear how the assumption of the normal distribution is confirmed, and non-normal distribution data is transformed to apply a parametric statistical test. Third, results from comparisons between the serum of patients are not presented; therefore, it is unclear how one-way ANOVA was used to analyze the relationship between GPC-3 domains in HCC and healthy patients, as described in the Statistical Analysis section. Also, one-way ANOVA was applied to compare the proteolytic activity of Furin-convertase (Fig. 3) but did not describe how multiple comparisons were tested. In addition, the authors should confirm if a post hoc analysis was conducted. Lastly, the statistical values of each experiment are desirable to include in the text or reported as supplementary data to the manuscript, such as exact p-values, error values, and confidence intervals. The authors are encouraged to provide more information on these points to improve the reliability of their findings.

Answer: Thank you. Sorry for the poor description of statistical analysis. We improved the statistical analysis section as requested.

Another area of concern is validity and interpreting the results.

-          The results from the first experiment reported in Fig.2a-c seem to be essential to understand why the authors continue investigations primarily on cirrhotic proximal (CP) samples in comparison to normal (NL) hepatocytes or cirrhotic distal (CD) or HCC cells. Otherwise, the authors not described or interpret these results in the manuscript and are only reported in Fig. 2. The following results to evaluate the proteolytic activity of Furin were conducted only in CP-Hep (Fig. 3 and 4). 

Answer: Thank you for this important comment. We extended the reasons on why we use CP-Hep for further experimentations. We added description of Fig.2a-c and clarify the focus in CP-Hep in Section 2.2

-          Complete information on antibodies should be fully described in the Methods. This is particularly important to understand how GPC-3 fraction N-terminal and C-terminal results are compared as one of the most relevant results of this study.

Answer: Thank you. We included that information in Method section “Protein Extraction and Western blotting”.

-          In this regard, another technique or statistical analysis should confirm the specificity of GPC-3 C-terminal expression in microvesicles. The authors proposed that the Pearson coefficient confirms the increase in the expression of C-terminal GPC-3. However, this statistical approach only demonstrates the linear relationship between two variables, in this case, the expression of GPC-3 and CD63 (a specific marker of microvesicles). Considering that this co-expression value does not confirm colocalization between two features is important. From the fluorescence images obtained, the authors could evaluate the relative fluorescence intensity resulting from colocalization (yellow) of GPC-3 (green) and CD63 (red); these results should be analyzed by Manders’ Overlap coefficient, taking into account the overlap between the two channels of fluorescence and provide a more accurate assessment of colocalization. Also, relative protein levels could be measured and compared between cytosol and purified microvesicles. The blots in Figure 2 suggest that a similar analysis was done but do not clearly show that this evaluation was performed. Fig. 2f and 2g show similar results, but only the expression of GPC-3 in microvesicles and GPC-3 N-terminal and C-terminal protein levels seem similar in 1st passage. Also, is confusing the results plotted in Fig.2h-j; it is not clear if the results are from the 1st or 8th passages and why this is relevant to this study. Given the relevance of this paper in discussing functional differences in N- and C-terminal fractions of GPC-3, authors should consider validating their results using another method, such as co-immunoprecipitation or proximity ligation assays.

Answer: Thank you for these comments. We include the Manders’ Overlap Coefficient to provide a better assessment of colocalization in Fig. 2e.

Sorry for the confusion. We replaced quantification in Fig. 2h we included EVs quantification of C- and N-terminal domains of GPC3, confirming the larger amount of C-terminal domain in EVs.

We also upgrade graphs in Fig2I, j. The results from early and late-stage cultures is expressed in the graph just to confirm expression of GPC-3 domains in small EVs.

Thank you for the technical suggestions. Soon upon funds availability analysis will be performed considering co-IP and PLA assays suggested by the reviewer.

-          Extracellular vesicles (EVs) and Microvesicles (MVs) terms are arbitrarily exchanged, which generates confusion on results. i.e., Figure 2f-g refers to MVs, but Fig.2k refers to EVs. According to the figure legend, these are the same results from hepatocytes. Therefore, it is necessary to adapt the terminology according to the current guidelines of the International Society of Extracellular Vesicles.

Answer: Thank you for these important comments. Following the nomenclature guidelines of MISEV20181,2 we use the term: small extracellular vesicles (sEVs) in the manuscript. Based on size nomenclature as particles <200nm in size. We adjust accordingly the title, main text and the figures labeling.

  1. Clotilde Théry et al. Minimal information for studies of extracellular vesicles 2018 (MISEV2018): a position statement of the International Society for Extracellular Vesicles and update of the MISEV2014 guidelines, Journal of Extracellular Vesicles, 7:1, DOI: 10.1080/20013078.2018.1535750
  2. MISEV – International Society of Extracellular Vesicles website.

-          It is hard to understand the relevance of measuring the area and volume of cells GPC-3 N-terminal positive cells to the aim of the study. According to the previous results reported in this manuscript, GPC-3 C-terminal is the cleaved form present in microvesicles from CP-hepatocytes. It seems not to have (or not been discussed) a relationship with the altered area and volume of cells by Furin inhibition with CMK. As no relationship is probed between the inhibition of GPC-3 cleavage, its expression in microvesicles and its relation to the reported disturbances has not been established; it does not seem relevant to evaluate GPC-3c levels from serum vesicles as a possible biomarker for HCC.

Answer: Thank you. Sorry, this was misleading. The cytoplasmatic GPC-3N 3D analysis and images were reported to support the cytoplasmic shrinkage that hepatocytes undergo after exposure to CMK. No correlation in the present study is reported between inhibition of the cleavage, presence of GPC-3 in the EVs as noticed by the reviewer. 

-          In this sense, results from serum microvesicles in patients should be expanded to arrive at a more objective interpretation. The sample of 3 control subjects should be increased, and samples from cirrhotic patients would be included according to their previous results. Also, it is not clear the difference from immunoblots in Fig. 5c, 5d, and 5e; it seems that blots from 5c and 5d are the same, and differences should be reported quantitatively as in their previous results. In addition, blots from 5e are different in levels of protein of GPC-3 N-terminal. Lastly, if GPC-3 N-terminal is only expressed in the cytoplasm (Fig. 3C), this is an inconsistent result, whereas, in serum-purified microvesicles from patients, GPC-3 N-terminal is expressed (Fig. 5f).

Answer: Thank you for these comments. We agree with the reviewer on the limited number of control case. We included a limitation section where we introduce the limitations to have few control subjects, not many patients with healthy liver go under surgery, so the limited number is due to the few procedures connected to gallbladder removal surgery.

Immunoblots in Fig.5d is the same blot in Fig. 5c but it is the merge between GPC-3 N- and C-terminal immunoblot. In Fig. 5d is the GPC-3N IB represented the black/white to show better the bands pattern of GPC-3N IB. We clarified this in Fig.5 legend. Fig. 5e is a schematic not an IB.

Thank you. This is a very important observation. The presence of GPC-3N in serum sEVs from HCC patient could indicate changes in its presence during in-vitro conditions of hepatocytes. More variegated EVs population are collected in the serum compared to those collected from culture media of primary hepatocytes. Moreover, GPC-3N domain is overlapping with CD63 (Fig. 2d) within hepatocytes showing very low level in IB in Fig. 3C.

Considering the authors’ need to review and confirm their results, their discussion should be expanded and contrasted with previous studies.

-          i.e., when the authors described that regulation of post-translational modifications was not evaluated in this study, authors should discuss if mechanisms that regulate GPC-3 post-translational changes were previously reported.

-          Authors should discuss why this post-translational modification described in HCC patients explains a physiopathological process associated with cancer progression. Without this, it is hard to understand how this finding could improve HCC diagnosis and prognosis.

Answer: Thank you we extended the impact of GPC-3 PTMs in the discussion.

-          Another point to discuss is what is previously reported about N- and C- cleaved forms of GPC-3 on target tissues in the context of cancer signaling mediated by microvesicles.

Answer: Thank you for this comment. We expanded this in the discussion.

Finally, it is inaccurate to assume these results support the idea that Furin-dependent GPC-3 cleaved domains could be a powerful tool for detecting initial stages. Therefore, authors should be focused on validating the expression of cleaved forms in cell lines derived from HCC patients and different clinical conditions (cirrhotic and non-cirrhotic patients and different stages of HCC) to assure the validity of GPC-3 cleaved forms in the diagnosis and prognosis of HCC.

Answer: Thank you for this comment. We observed in published data1 that Furin enzyme is overexpressed in late passages in CD- and CP-Hep cultures, at the same time we observed an increment of GPC-3 domains formation. These data suggested at that time that increment of Furin levels and activity produced more GPC-3 cleavage forms. We still do not know the meaning and function of these forms in the neoplastic context. Soon we hope to investigate the effects of expression/overexpression of GPC-3 forms as diagnostic and prognostic markers alongside molecular study to investigate specific functions of GPC-3 forms in hepatocytes during their neoplastic transformation.

  1. Montalbano, M., Rastellini, C., McGuire, J.T. et al.Role of Glypican-3 in the growth, migration and invasion of primary hepatocytes isolated from patients with hepatocellular carcinoma. Cell Oncol. 41, 169–184 (2018). https://doi.org/10.1007/s13402-017-0364-2

There are also minor comments that the authors should consider:

-          Citation style is not consistent (numbers than full cite)

Thank you. Corrected

-          There is information that needs to be properly cited (i.e., lines 56-57, 63-67, 69, 77-79)

Thank you. Corrected

-          ‘It is not surprising does not seem a sufficiently substantiated argument to suggest that GPC-3 could be part of a signaling mechanism mediated by microvesicles (Line 71-72)

Answer: we agree with the reviewer no evidence is published on signaling functions of cleaved forms of GPC-3. We did not speculate on possible function in known pathways where GPC-3 is involved but just reported what has been observed in previous studies by others.

-          Several references are duplicated.

Answer. Thank you corrected.

-          Abbreviatures are first described in the Methods section; this may not be functional because this is the last section of the manuscript.

Answer: Thank you. We corrected it.

-          Figures are not correctly cited in the text. (i.e., line 134 describes the N-terminal of GPC-3 detected on cytosol and membrane of hepatocytes, referencing Fig. 2A. However, Fig. 2A is a schematic view of the multichambered system used to evaluate migration).

Answer: Thank you and sorry for the confusion we correct the citation of figures. Also, we improved description of results represented in Fig.2.

Overall, there are major areas to review in statistical methods, the validity and interpretation of results, discussion and citing, and minor comments about attending in general in the manuscript to conclude that on this initial state, the manuscript should be rejected to help the authors to produce a more impactful and reliable study.

Answer: We thank one more time the reviewer for his/her objective and deep critiques we appreciate the suggestions proposed to improve the manuscript. We strongly believe following reviewer suggestions that now the manuscript is in a better shape and suitable for publication in IJMS.

Round 2

Reviewer 1 Report

This revised version is included in what I request. I have nothing further to point out.

Author Response

We thank the reviewer for the revision. we believe now the manuscript is improved in quality and shape.

Reviewer 2 Report

The study by Mauro et al. was improved, and several points were clarified to respond to the previous comments. There are remaining points to be addressed before considering the article for publication. Most of them are in the same line as the earlier comments, focusing on the major concerns unresolved. I have provided additional comments that need attention.

I look forward to reviewing the revised manuscript.

1.                 The section on statistical analysis was improved. In addition, as may seem obvious, significance levels would be described in text or figure. In this case, if the authors already illustrated in Figure 3 (****, p<0.001), significance levels should be added to Figures 1 & 4.

2.                 The authors improve the validity and interpretation of the results. However, to consider resolving the points remarked in Figure 2, authors should improve the quality of the graphs (in this graph, and in general in all figures) because the level of significance (*) is not distinguishable from non-significance (ns). For example, the only result that seems to be significant in Fig. 2C is the percentage of area covered in NL-Hep/CD-Hep between 6h and 12h. This contradicts what is described in the text, where the authors claim, “We observed a significant increase in the number of migrating NL-Hep induced by CP-Hep at 24h” (Line 163-164). If this statement is not supported and clearly described in Figure 2, it is still unclear why to choose CP-Hep for the following experiments.

3.                 Manders’ Overlap coefficient needs to be described in the Methods. Also, Figure 2e stills need to be updated because it already represents the same previous Pearson coefficient. Results also be included and described in the text. At this point, there is no evidence this analysis was incorporated.

4.                 Fig. 2f-k are the same as the previous version. As no update has been made to this section, the below comment is considered unresolved, and the authors are invited to view the comments.

-          Previous comment: The blots in Figure 2 suggest that a similar analysis was done but do not clearly show that this evaluation was performed. Fig. 2f and 2g show similar results, but only the expression of GPC-3 in microvesicles and GPC-3 N-terminal and C-terminal protein levels seem similar in 1st passage. Also, is confusing the results plotted in Fig.2h-j; it is not clear if the results are from the 1st or 8th passages and why this is relevant to this study.

5.                 Nomenclature also needs to be updated in Figures. Fig.2,3, and 5 still use the term MVs or EVs instead of sEVs.

6.                 The comment regarding the relevance of measuring the area and volume of cells GPC-3 N-terminal positive cells to the aim of the study is still going in the same direction. These results support cytoplasmic shrinkage observed in Fig.3. However, these results are still irrelevant to this study’s aim and should be included as supplementary results. Therefore, these results are included in section 2.3, where the objective is to describe ‘GPC-3 C-terminal domain product from proteolytic Furin-Convertase induced cleavage is localized in hepatocyte small EVs’. In contradiction to this objective, it is more interesting to study the effects of the cleaved GPC-3 N-terminal form on the increase in anoikis and cell shrinkage. In this regard, it is not clear what is the relevance of knowing that the C-terminal form is expressed in sEVs.

7.                 The differences in blots from Fig.5c-d are clarified. However, it is still unclear why the blots from Fig.5f differ. This is because both GPC-3 levels in Fig5c-d and GPC-3 in Fig5f are from serum-purified sEVs in healthy and HCC patients, but the band's levels and shape are different.

8.                 There are still minor issues in the citation; in line 40, Curcuru’?? is the correct name from the author? Other issues identified:  Some authors are in uppercase; more than one author name is cited in the text instead of et al.; first and second name author in the text, following et al.; missing cites in Discussion: Line 315-316, 316-317, 318-321.

Overall, the manuscript was improved substantially, but the comments above must be addressed for publication.

Author Response

Comments and Suggestions for Authors

The study by Mauro et al. was improved, and several points were clarified to respond to the previous comments. There are remaining points to be addressed before considering the article for publication. Most of them are in the same line as the earlier comments, focusing on the major concerns unresolved. I have provided additional comments that need attention.

I look forward to reviewing the revised manuscript.

We thank the reviewer for these additional comments. We addressed each criticism; we believe now the manuscript is improved in quality and shape. We also substituted the old figures in the main text.

  1. The section on statistical analysis was improved. In addition, as may seem obvious, significance levels would be described in text or figure. In this case, if the authors already illustrated in Figure 3 (****, p<0.001), significance levels should be added to Figures 1 & 4.

 Thank you. We added missing significance information in Figure legend 2, 3 and 4.

  1. The authors improve the validity and interpretation of the results. However, to consider resolving the points remarked in Figure 2, authors should improve the quality of the graphs (in this graph, and in general in all figures) because the level of significance (*) is not distinguishable from non-significance (ns). For example, the only result that seems to be significant in Fig. 2C is the percentage of area covered in NL-Hep/CD-Hep between 6h and 12h. This contradicts what is described in the text, where the authors claim, “We observed a significant increase in the number of migrating NL-Hep induced by CP-Hep at 24h” (Line 163-164). If this statement is not supported and clearly described in Figure 2, it is still unclear why to choose CP-Hep for the following experiments.

 Thank you. We corrected Figure 2c statement and report (implementing the analysis) significant change at 24h in NL-Hep/CP-Hep co-culture.

  1. Manders’ Overlap coefficient needs to be described in the Methods. Also, Figure 2e stills need to be updated because it already represents the same previous Pearson coefficient. Results also be included and described in the text. At this point, there is no evidence this analysis was incorporated.

 Thank you. We believe you had access to the old figures, sorry for the inconvenience. We incorporated new Figure 2e, data description and method in the manuscript.

  1. Fig. 2f-k are the same as the previous version. As no update has been made to this section, the below comment is considered unresolved, and the authors are invited to view the comments.

-          Previous comment: The blots in Figure 2 suggest that a similar analysis was done but do not clearly show that this evaluation was performed. Fig. 2f and 2g show similar results, but only the expression of GPC-3 in microvesicles and GPC-3 N-terminal and C-terminal protein levels seem similar in 1st passage. Also, is confusing the results plotted in Fig.2h-j; it is not clear if the results are from the 1st or 8th passages and why this is relevant to this study.

Thank you. Sorry for the confusion. To clarify we update Fig. 2h quantification. In 2h, we compare GPC-3 C and N domains in small EVs content (Immunoblot in 2f and 2g, respectively), confirming higher levels of C-terminal domain than N-terminal. Moreover, we include two new graphs one for GPC-3N and one for GPC-3C to visualize change at the 1st and 8th culture passages in small EVs content. 

  1. Nomenclature also needs to be updated in Figures. Fig.2,3, and 5 still use the term MVs or EVs instead of sEVs.

Thank you corrected.

  1. The comment regarding the relevance of measuring the area and volume of cells GPC-3 N-terminal positive cells to the aim of the study is still going in the same direction. These results support cytoplasmic shrinkage observed in Fig.3. However, these results are still irrelevant to this study’s aim and should be included as supplementary results. Therefore, these results are included in section 2.3, where the objective is to describe ‘GPC-3 C-terminal domain product from proteolytic Furin-Convertase induced cleavage is localized in hepatocyte small EVs’. In contradiction to this objective, it is more interesting to study the effects of the cleaved GPC-3 N-terminal form on the increase in anoikis and cell shrinkage. In this regard, it is not clear what is the relevance of knowing that the C-terminal form is expressed in sEVs.

Thank you. We agree with the reviewer comment and rearranged Figure 4 as Supplemental Fig. 2. So Figure 5 become Figure 4.

  1. The differences in blots from Fig.5c-d are clarified. However, it is still unclear why the blots from Fig.5f differ. This is because both GPC-3 levels in Fig5c-d and GPC-3 in Fig5f are from serum-purified sEVs in healthy and HCC patients, but the band's levels and shape are different.

Thank you. We included molecular weight markers to clarify band location. The bands in 5f present a smear (mainly in 50 kDa form) present in 5c. The shape looks more swollen because the SDS-PAGE running and the low/absence amount of proteins (at the same molecular weight) in the nearby Supernatant samples. 

  1. There are still minor issues in the citation; in line 40, Curcuru’?? is the correct name from the author? Other issues identified:  Some authors are in uppercase; more than one author name is cited in the text instead of et al.; first and second name author in the text, following et al.; missing cites in Discussion: Line 315-316, 316-317, 318-321.

Thank you. We corrected with Curcuru`. Formatting of reference is automatic from the template. We added the missing references.

Overall, the manuscript was improved substantially, but the comments above must be addressed for publication.

We would like to thank the reviewer for the comments, which improved substantially the quality of this manuscript.

Round 3

Reviewer 2 Report

Comments and Suggestions for Authors

The study by Mauro et al. was improved, and most comments have been addressed or clarified. After reviewing, authors should attend to minor corrections missing from the previous suggestions. There is no need to return the manuscript to follow the revision. After attending to the minor changes, the article may be considered for publication.

1.       Improve the quality of Figure 1

2.       Include molecular weight markers in Fig. 3C and Fig. 4f. Also, the differences in GPC-3N in sEVs and supernatant from serum patients (Fig.4f) should be described in the results.

3.       Provide the same terminology to refer to the comparison between cells, by passages or by weeks of culture.

4.       The text and figures provide the same terminology: Manders’ overlap coefficient (MOC) or Manders’ colocalization coefficient (MCC).

5.       Provide the same terminology to refer to small extracellular vesicles in Supplementary figures.

Author Response

The study by Mauro et al. was improved, and most comments have been addressed or clarified. After reviewing, authors should attend to minor corrections missing from the previous suggestions. There is no need to return the manuscript to follow the revision. After attending to the minor changes, the article may be considered for publication.

  1. Improve the quality of Figure 1

Thank you. We place a better resolved Figure 1.

  1. Include molecular weight markers in Fig. 3C and Fig. 4f. Also, the differences in GPC-3N in sEVs and supernatant from serum patients (Fig.4f) should be described in the results.

Thank you. Included missing weight markers. Description has been included Line-288-291

  1. Provide the same terminology to refer to the comparison between cells, by passages or by weeks of culture.

Thank you. Corrected.

  1. The text and figures provide the same terminology: Manders’ overlap coefficient (MOC) or Manders’ colocalization coefficient (MCC).

Thank you. Corrected.

  1. Provide the same terminology to refer to small extracellular vesicles in Supplementary figures.

Thank you. Corrected in Supplementary Figure 1.